# Innovative Activated Carbon Based on Deep Eutectic Solvents (DES) and H$_3$PO$_4$

**Aloysius Akaangee Pam**

Chemistry Department, Faculty of Science, Federal University Lokoja, P.M.B. 1154 Lokoja, Nigeria;
aloysius.pam@fulokoja.edu.ng

**Abstract:** In this present work, a novel method for synthesis of palm kernel shell activated carbon was established using DES (choline chloride/urea)/H$_3$PO$_4$ as the activating agent. The pore characterization, morphology, and adsorption properties of the activated carbons were investigated. The activated carbon samples made from the same feedstock at two pyrolysis temperatures (500 and 600 °C) were compared for their ability to adsorb Pb(II) in aqueous solution. The results demonstrated that the production of the activated carbon and adsorptive properties were significantly influenced by the pyrolysis temperature and the ratio of precursor to activating agent. DES/H$_3$PO$_4$ activated carbon (having surface area 1413 m$^2$/g and total pore volume 0.6181 cm$^3$/g) demonstrated good Pb(II) removal. Although all the tested activated carbon samples adsorbed Pb(II) from aqueous solution, they demonstrated different adsorption capabilities according to their various properties. The pyrolysis temperature, however, showed little influence on the activated carbon adsorption of Pb(II) when compared to the impregnation ratio. Their good desorption performance perhaps resulted from the porous structure.

**Keywords:** Activated carbon; palm kernel shell; lead; surface area; total pore volume

## 1. Introduction

Activated carbon, a highly multi-porous structure material with large developed surface area, is without doubt an amazing material that has been widely explored either on its own or as a composite material for many decades. Owing to the fascinating structural properties, activated carbon is suitable for various applications, including water purification [1], stem cell growth [2], catalyst support [3], storage, separation and purification of gases [3,4], drug delivery [5], and as electrodes of super capacitors. Since most of the activated carbons are pyrogenic carbonaceous materials, they are prepared by thermochemical conversions and the use of some chemicals such as NaOH, ZnCl, can lead to secondary pollution. In order to avoid these environmental concerns, the design of novel materials and the development of synthesis routes to improve the structural and chemical characteristics of the material, in addition to seeking greener and economical material synthesis methods, were undertaken. Deep eutectic solvents (DESs) are promising sustainable options to conventional solvents and ionic liquids [6], owing to the fact that research into ionic liquid (IL) analogues (DESs) has increased in recent years [7]. DESs are systems formed from a eutectic mixture of Lewis or Brønsted acids and bases, which can contain a variety of anionic and/or cationic species [8]. DES, including a mixture of choline chloride and urea in a molar ratio of 1:2 with a melting point of around 12 °C is another form of ionic solvent [9]. DESs are known for their low toxicity, high biodegradability, high recyclability, high thermal stability, low inflammability, and low volatility [6]. The diversity and intriguing physical properties of DES are manifested in a wide range of potential applications such as polymer synthesis, organic reactions, sustainable solvents, green chemistry, electrochemistry, enzyme reactions, biodiesel treatments, synthesis of molecular sieves, zeolites, porous materials [10], and developments

of nanomaterials for environmental applications as well as many others. Recently, various materials have been used with DES to synthesize different composites for the removal of various pollutants in aqueous solution [11–13]. In addition, many studies have shown the possibility and efficiency of using DESs as functionalization agents to modify carbon nanotubes nanostructure [14,15], graphene, and graphene oxide [16,17], and Carbon Monoliths [18]. Recently also, the impregnation of activated carbon (which has shown great potential for the removal of different types of pollutants) with DES and other chemicals has resulted in control of the pore size and pore surface chemistry of the resulting carbon and a higher amount of pollutant removal. Herein, a novel concept is proposed for the synthesis of activated carbon using palm kernel shell (PKS) in DES (choline chloride/urea)/ $H_3PO_4$ for the first time. The choice of $H_3PO_4$ is because of the relatively lower, environmental and toxicological constrains compared to $ZnCl_2$, and lower working temperature compared to KOH or NaOH [19]. In this work, PKS was converted to high surface area carbons by treating with DES/$H_3PO_4$ at different ratios by weight and then heat-treating the mixture at temperatures ranging from 500–600 °C. The effect of activating agent on surface area, surface morphology and Pb(II) adsorption properties were investigated. The obtained carbon materials were characterized by Scanning Electron Microscope (SEM), and Fourier Transform-infrared (FT-IR). Surface area and pore volume were determined by nitrogen/desorption measurements. A comparably high specific surface of 1413 $m^2$/g and pore volume of 0.598 $cm^3$/g was achieved when the activated carbons (ACs) were prepared with an activating agent (DES/$H_3PO_4$) to PKS ratio of 2:3 (g/g) and activating temperature 600 °C for 2 h. The values of the pore structure are much higher than obtained in other agricultural materials and commercially available activated carbon, indicating that porous carbon material derived from PKS using DES/$H_3PO_4$ as an activating agent would have potential for wastewater treatment.

## 2. Materials and Methods

### 2.1. Reagents

The main chemicals for this study were purchased from Sigma-Aldrich Company (St. Louis, MO, USA), including choline chloride and urea. All other used chemicals were of high purity. All aqueous solutions were prepared using deionized (DI) water. The preparation of Pb(II) stock solution (1000 mg/L), was by dissolving the appropriate amount of lead(II) nitrate (Pb(NO$_3$)$_2$) (Fisher Scientific, Pittsburgh, PA, USA) in ultrapure water. The working solutions were prepared by the dilution of the stock solutions to 50 mg/L.

### 2.2. Synthesis of Choline Chloride–Urea Deep Eutectic Solvent

The used DES was synthesized according to a reported method [20]. This involved reaction of choline chloride (from Sigma-Aldrich) and urea (from Sigma-Aldrich) in a mole ratio of 1:2 and with stirring under heating (80 °C) afforded a homogeneous, clear solution. The DES (chcl-2urea) was then used for the preparation of activated carbon without further purification.

### 2.3. Preparation of Activated Carbon

The activated carbon (AC) was prepared using palm kernel. The choline chloride (chcl)/urea deep eutectic solvents were mixed with $H_3PO_4$ in the ratio (1:4 and 2:3 g/g) and stirred for about 3 h. The ratio of (choline chloride + urea)/$H_3PO_4$ to palm kernel shell was 2:1. The sample was then heated at a fixed rate of approximately 10 °C/min to 500 and 600 °C, respectively, and held at these temperatures for 2 h. The ACs obtained were thoroughly washed with hot distilled water several times, oven dried at 100 °C overnight, cooled to room temperature. The prepared activated carbon



was ground and passed through a 200-mesh sieve, then stored in sample bottles for characterization. The activated carbon yield was calculated according to Equation (1).

$$Yield(\%) = \frac{Mass\ of\ sample\ after\ activation(g)}{Initial\ mass\ of\ dried\ sample(g)} \quad (1)$$

*2.4. Characterization of Adsorbents*

Specific Surface Area and Pore-Distribution

The BET surface area ($S_{BET}$) and pore-size distribution of AC-600 2:3, AC-600 1:4, and AC-500 2:3 were obtained by Quanta chrome autosorb automated gas sorption instrument (Boynton Beach, FL, USA) using sorption of nitrogen at 77 K. Prior to analysis, the adsorbents were out-gassed for 12 h under vacuum at 110 °C. The specific surface area was determined according to the BET method at the relative pressure at 33.5 atm. $S_{BET}$ was calculated in the range of relative pressure from 0.025 to 0.3 [21]. The pore size distribution functions (PSDs) were calculated from the adsorption branch of isotherms using the Kruk–Jaroniec-Sayari (KJS) method based on the Barrett–Joyner–Halenda (BJH) calculation procedure for cylindrical pores [22]

*2.5. Functional Groups Determination*

To investigate the surface chemistry, the infrared spectrum of the ACs was recorded on a Perkin-Elmer spectrum 100 (Shelton, CT, USA), spectrophotometer using attenuated total reflectance sampling techniques. The samples were dried to a constant weight, the ground samples (approximately 20 mg) were placed on top of the attenuated total reflection (ATR) crystal using a mechanical anvil, and spectra were collected between 4000 and 400 cm$^{-1}$ at a resolution of 4 cm$^{-1}$ averaging 32 scans per sample.

*2.6. Morphological Characterization*

The elemental composition was investigated using energy dispersive X-ray spectroscopy (DAX TEAM, Ametek, Bereyn, PA, USA), integrated with high-resolution FEI Nova 230 Field Emission Scanning Electron Microscope (FESEM, Denton, TX, USA). Prior to imaging, activated carbon samples were attached on self-adhesive carbon sticky tape and gold-coated using auto fine coater (JFC-1600, Akishima, Japan) operated at 20 mA for 140 s to prevent charging.

## 3. Results and Discussion

*3.1. Fourier Transform Infrared Spectroscopy*

Fourier transform infrared spectroscopy (FT-IR, Boynton Beach, FL, USA) analysis of the ACs is presented in Figure 1. The spectra of these samples(AC-600 2:3, AC-600 1:4,AC-500 2:3, and AC-500 1:4) present peaks in the region 1158, 3248, 2900, 1742, and 1061 cm$^{-1}$ corresponding to the P=O bond in phosphate ester, O–C bond (in the P–O–C linkage or P=OOH bond), OH stretch, CH, C=O, C–O [23–25], respectively. FT-IR spectra showed almost a similar pattern and the functional groups assignment due to the same precursor were used. There were no significant changes observed in spectra when varying the concentration of dehydrating agent. Similar spectra results have also been obtained for AC prepared from rice husk [26].

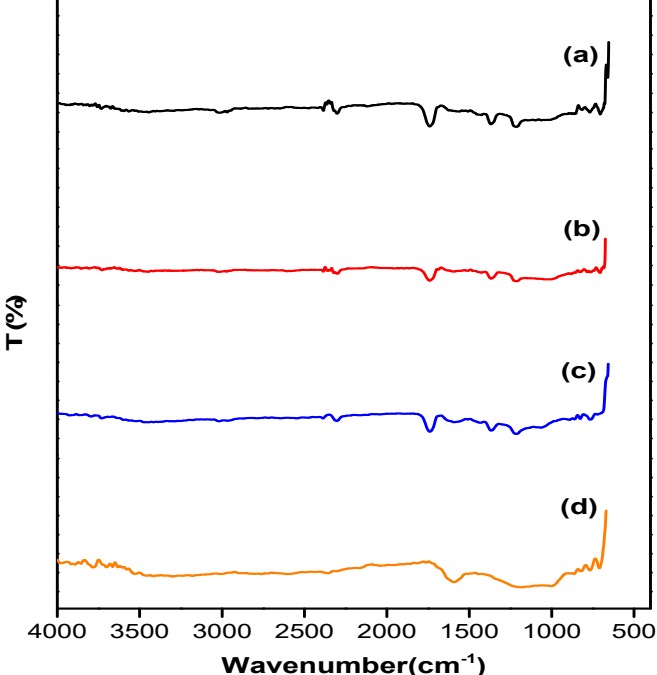

**Figure 1.** Fourier transform infrared spectroscopy of (**a**) AC-600 2:3; (**b**) AC-600 1:4; (**c**) AC-500 2:3 and (**d**) AC-500 1:4.

### 3.2. Surface Morphology

The surface morphology of the different adsorbents observed with SEM using the same magnification (5000×) are shown in Figure 2a–d. Evidently, the pore volume was very limited and not well developed for AC-500 2:3. However, after activation at 600 °C, irrespective of the impregnation ratio, a large number of cavities developed (Figure 2c,d), as confirmed by the BET results (Table 1). The development of the pore was attributed to the decomposition of the lignocellulosic structure in the precursor due to thermal expansion during the pyrolysis process while the use of DES/$H_3PO_4$ activating agent helped in the pore development. Moreover, the presence of $H_3PO_4$ promotes pyrolytic dehydration, and decomposition of the constituent lignocellulosic material and formation of a cross-linked structure [27], during the pyrolysis process. This was reasonable enough as increasing the ratio DES/$H_3PO_4$ would have not only have activated the carbon material but would have removed the volatiles from the biopolymer more completely, in turn supporting the much improved pore volume and surface area observed. A few pores in the sample seems to have been clogged by the DES/$H_3PO_4$ especially as in Figure 2c,d.

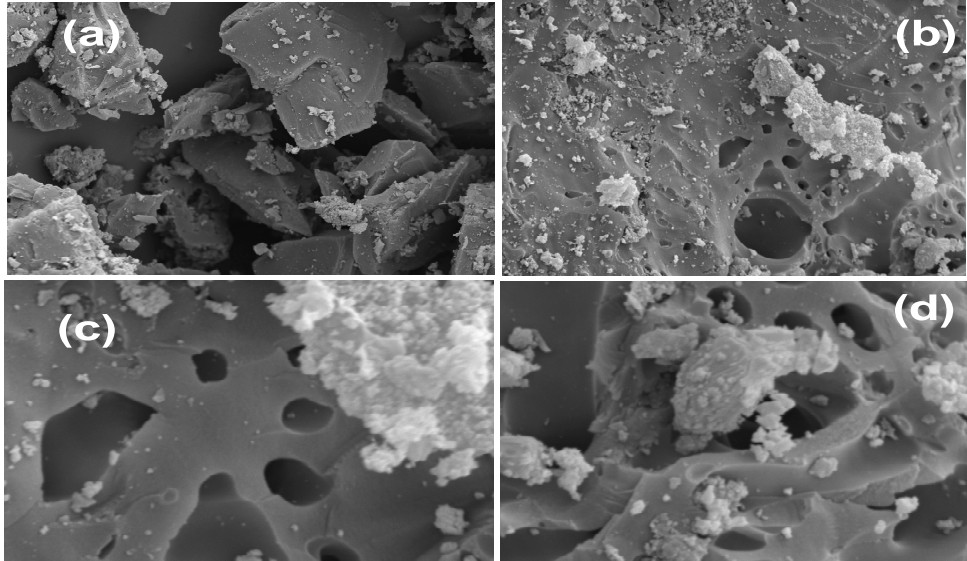

**Figure 2.** SEM micrograph of prepared adsorbents (**a**) AC-500 2:3; (**b**) AC-500 1:4; (**c**) AC-600 2:3; (**d**) AC-600 1:4.

**Table 1.** Surface area and porosity of activated carbon.

| Sample | Yield (%) | $S_{BET}$ (m²/g) | Micropore Volume ($V_{mic}$, cm³/g) | Total Pore Volume (cm³/g) | Average Pore Width (Å) |
|---|---|---|---|---|---|
| AC-600 2:3 | 30.1 | 1413 | 0.5980 | 0.6181 | 11.002 |
| AC-600 1:4 | 30.7 | 154.0 | 0.2850 | 0.3517 | 5.4697 |
| AC-500 2:3 | 55.5 | 337.2 | 0.3726 | 0. 3921 | 5.5264 |

### 3.3. BET Surface Area and Pore Characteristics

Table 1 shows the BET surface area and the effect of the activation temperature on the carbon yield. The results show that the yield decreased from 55.5% to 30.7% when the activation temperature was increased from 500 to 600 °C. This is because the higher activation temperature released more volatiles and consequently reduced the carbon yield. As the activation temperature was increased from 500 to 600 °C, the surface area decreased from 1413 to 337.2 m²/g. However, further increases in the DES/$H_3PO_4$ ratio (from 2:3) led to a marked increase in both carbon surface area and pore volume due to the micropores having been widened at the same temperature. The maximum BET surface area of 1413 m² /g was obtained when the ratio of DES:$H_3PO_4$ was 2:3 g/g. A similar trend was also reported by Yorgun et al. [27] in their preparation of activated carbon from paulownia wood with $H_3PO_4$. In addition, at a low ratio of DES/$H_3PO_4$, the proportion of the micropore volume decreased up to 45.5%. The nitrogen adsorption isotherms and porous structure parameters was studied for AC-600 2:3, AC-600 1:4, AC-500 2:3 samples alone, because of their high porosity (Figure 2). Figure 3a shows the nitrogen adsorption isotherm of AC-600 2:3, AC-600 1:4 and AC-5002:3. According to the IUPAC classification, the isotherm belongs to type I, typical of microporous materials. The average pore diameters were between 6.0 and 12.0 Å (BJH method), and the pore size distribution concentrated around 8.0 to 10.20 Å (Figure 3b–d), which showed that all the adsorbent was microporous in nature. With the increase of the DES/$H_3PO_4$ content, BET, average pore diameter, and $V_{mic}$ all increased.

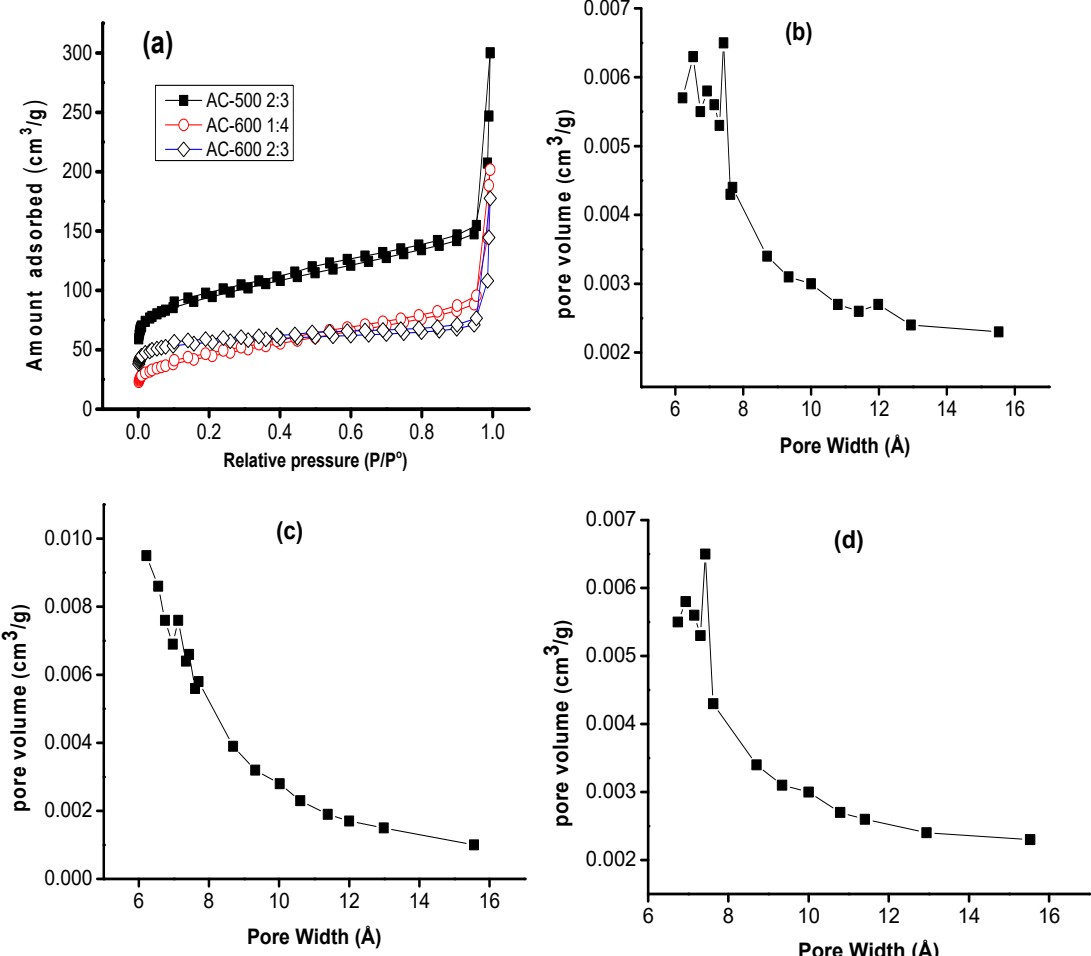

**Figure 3.** (**a**) Adsorption–desorption isotherms of N$_2$ and (**b**), (**c**), and (**d**) are the corresponding pore size distributions curves as obtained by the BJH method for AC-600 2:3, AC-600 1:4 and AC-500 2:3, respectively.

The surface textural properties (pore volume and surface area) of the prepared AC at optimum conditions were compared with similar carbon materials obtained from different agricultural waste and commercial activated carbon. As can be seen in Table 2, the estimated $S_{\text{BET}}$ (m$^2$/g), 1413 m$^2$/g is considered high and it is in the range of commercial biomass-based derived carbon (500–1500 m$^2$/g) [28]. The highest total pore volume for the AC is 0.6181 cm$^3$/g which is also larger than those of commercial activated carbon, i.e. 0.172, 0.369, and 0.250 cm$^3$/g [29], 0.60 and 0.52 cm$^3$/g for BPL and PCB, respectively, (Calgon Carbon Co., Pittsburgh, PA, USA) [30]. Phosphoric acid has been reported to produce an activated carbon high degree of intrinsic pore structure with high percentage of micropores and mesopores [31]. Moreover, it has been reported that DES can assist in the formation of pores during the synthesis of porous carbon material [32]. These factors perhaps could be responsible for the good microstructure activated carbon (with acceptable quality for surface area and total pore volume) reported in this work.


**Table 2.** The surface textural properties of the prepared active carbons (ACs) were compared with similar carbon materials obtained from different agricultural waste.

| Precursor | Activating Agent | Temperature (°C) | $S_{BET}$ (m²/g) | Total Pore Volume (cm³/g) | References |
|---|---|---|---|---|---|
| Palm kernel shell | DES/H$_3$PO$_4$ | 600 | 1413 | 0.618 | This work |
| Rice husk | NaOH | 900 | 885 | 1.180 | [26] |
| Palm kernel shell | H$_3$PO$_4$ | 550–650 | 968–1153 | 0.677–0.711 | [33] |
| Hazelnut bagasse | KOH | 700 | 1642 | 0.964 | [34] |
| Hazelnut bagasse | ZnCl$_2$ | 600 | 1489 | 0.933 | [34] |
| Rice husk | | 700 | 750 | 0.380 | [21] |
| Bagasse | | 700 | 674 | 0.340 | [21] |
| Chinese fir sawdust | ZnCl$_2$ | 400–600 | 1071 | 0.566 | [35] |
| Green Coconut Shells | ZnCl$_2$ | 650 | 996 | 0.448 | [36] |
| CA | | | 648 | - | [37] |
| CA | | | 825 | 0.696 | [38] |
| coconut shell carbon, CA | | | 399 | 0.172 | [29] |
| bituminous carbon, CA | | | 961 | 0.369 | [29] |
| lignite carbon, CA | | | 658 | 0.250 | [29] |

CA = Commercial activated.

### 3.4. Adsorption of Lead

Motivated by the excellent pore structure of the activated carbons produced from palm kernel shell, their Pb(II) adsorption properties were investigated. The activated carbon with DES/H$_3$PO$_4$ (wt/wt) ratio was evaluated: AC-500 1:4, AC-500 2:3, AC-600 1:4, and AC-600 2:3. Figure 4 shows the adsorption capacities of various carbon materials for Pb(II). The adsorption capacities obtained from these adsorbents were 43.6 mg/g (AC-500 1:4) and 58.9 mg/g (AC-500 2:3), 47.5 mg/g (AC-600 1:4), and 58.1 mg/g (AC-600 2:3), for 50 mg/L of Pb(II) ions. Clearly, AC-600 2:3 showed a higher adsorption capacity for Pb(II) ions than that of the AC-500 1:4, AC-500 2:3 and AC-600 1:4. With the rise of the activation temperature, the adsorption capacity of ACs gradually increased particularly for the ACs derived at ratio DES/H$_3$PO$_4$ 2:3. This is an indication that the activation process apparently improved the adsorption capacity, which is consistent with the results of the specific surface area between AC-500 and AC-600. Moreover, the Pb(II) adsorption values of DES/H$_3$PO$_4$ activated carbon in this work was also much higher than that of rice husk by HCl Activation [39]. Besides, the yields of the ACs in this work were also comparable to that of other ACs from different precursors [27]. The excellent pore structure and larger average pore diameter (5.5–11.0 Å) of the ACs does permit Pb(II) particles of ionic radii 0.133 nm and easy diffusion into the pores [40]. Thus, DES/H$_3$PO$_4$–palm kernel activated carbon could be considered as a promising alternative for the preparation of activated carbon.

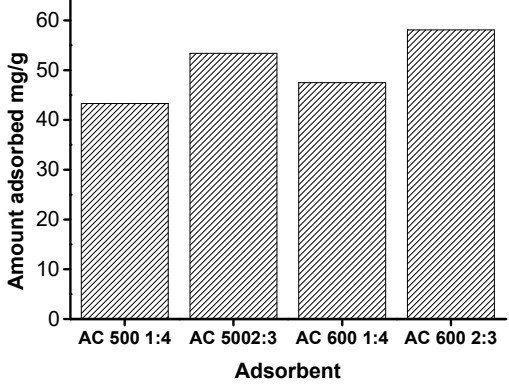

**Figure 4.** Pb(II) adsorption capacity of various carbon materials of AC-500 1:4, AC-500 2:3, AC-600 1:4, and AC-600 2:3 treated with 50 mg/L of Pb(II) (vol. = 200 mL; adsorption time = 2 h).

## 4. Conclusions

Using palm kernel shell, ACs comprising high surface area were successfully fabricated through a moderate pyrolysis temperature and DES/$H_3PO_4$ activation. Physicochemical properties of the resultant activated carbon (AC-600 2:3) revealed a large surface of 1413 $m^2$/g and total pore volume 0.6181 ($cm^3$/g) with narrow pore size distribution, which demonstrated more prospect for water treatment than the other activated carbons (500 1:4, AC-500 2:3, AC-600 1:4). Furthermore, the adsorption properties of the activated carbons in aqueous solution containing Pb(II) were also examined. The adsorption of AC-600 2:3, was higher than that of AC-500 1:4, AC-500 2:3, AC-600 1:4, respectively. In addition, the physicochemical properties of the ACs are better than other agricultural materials and commercially available activated carbons, demonstrating that the ACs prepared from environmental friendly activating agents and palm kernel shell would have potential as high efficiency biosorbents in water purification.

**Funding:** This research received no external funding.

**Conflicts of Interest:** I declare no conflict of interest.

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
