# Peer review of "Innovative Activated Carbon Based on Deep Eutectic Solvents (DES) and H3PO4"

_carbon_

Round 1
Reviewer 1 Report
The above-mentioned manuscript deals with a the preparation and characterization of activated carbons with large surface area and pore volumen to be employed to remove lead from aqueous solution.
I consider that the topic in the present paper is of interest from a practical application point of view. The content of the manuscript could be of interest for the readers, but the manuscript need some important modifications:
- There are many spelling errors through the manuscript, it is hard to read and understand the paper, such as: 500-6000C ( Line 57), incomplete equation (1), AC-6002 and AC-600 (Line 91), Figure 2 instead of Figure 3 (Line 154),....
- Introduction Section: the last part (lines 61 to 65) should not contain results of the work, only the aims of the study
- The data should be added in Table 1 for sample AC-500 1:4
- In the experimental section: What about the kinetic and adsorption experiments? , If adsorption results are provided, this information must be supplied.
- The difference and importance of this work compared with similar work should be clearly addressed. For example, use Table 2 for this discussion.
- How many replicates the author has carried out in each experimental adsorption measurement? That kind of information is mandatory from a scientific point of view.
- Finally, is the adsorption time (2h) the equilibrium time?, how do yo determined it?
Author Response
Please see attachment in the box

Reviewer 2 Report
The paper presents interesting issues. However, some remarks concerning the preparation of the manuscript should be taken into consideration while revising the paper:
- All references should be adjusted to the Materials rules. This applies to the references in both
the text and the References section.
- Page1, line 15, page 2, line 61, and page 7, line 193 are wrong units, correct is m2/ g.
- Page 2, line 57, incorrect temperature range 500-6000°C, should be 500-600°C.
- Page 3, line 91, not correctly presented symbols of studied coals, please correct.
- Page 3, line 90: 2.4.1. Specific surface area and pore- distribution; Please describe in more detail how calculations were carried out, e.g. as below:
SBET was calculated in the range of relative pressure from ……. to…..??? [references]
The pore size distribution functions (PSDs) were calculated……using the ………… method??? [references]
- Page 4, line 125 I do not agree with the conclusion that the materials tested have a honeycomb structure, it is hard to see such a structure in the SEM pictures presented.
- The following carbons were obtained in this work: AC-600 2:3, AC-600 1:4, AC-500 2:3, AC-500 1:4. Why the nitrogen adsorption isotherms and porous structure parameters for carbon AC-500 1:4 were not presented in the paper?
- Page 6, line 61; The volume of micropores (table1) was calculated in the work. What is the total pore volume? Please complete this parameter in table 1.
- Page 6, line 65;
Rice husk, NaOH, parameters: 900, 885, 1.18; (wrong literature reference [24], correct [23]. The value of 1.18 is the total volume pores, is compared with the volume of micropores of materials obtained in this work. I am asking for verification and clarification. In the paper [23] the authors also received other materials with significantly better parameters, why the table lists the parameters of this material?
Palm shell (wrong literature reference [26], please enter correct)
Hazelnut bagasse KOH and Hazelnut bagasse ZnCl2 (reference number [27] applies: “Chinese fir sawdust by zinc chloride activation under vacuum condition”; wrong literature reference [27], please enter correct.
Chinese fir sawdust The value of 0.5665 is the total volume pores, is compared with the volume of micropores of materials obtained in this work; wrong literature reference [28], please enter correct [27]
Green Coconut The value of 0.4487 is the total volume pores, is compared with the volume of micropores of materials obtained in this work; wrong literature reference [29], please enter correct [28]
Please, verify the parameters presented in the whole table (table 2) and check the literature, because references [30-32] are also incorrect.
- Please draw some conclusions based on the results presented in Table 2.
- Page 7, line 190; The paper presents only preliminary adsorption studies, and the conclusions state that these materials have excellent adsorption properties. The results obtained were not compared with those available in the literature. So why are the conclusions about the excellent adsorption properties of these materials?
- I think that the results of the research should be supplemented at work (fit to isotherm models, eg. Langmuir, Freundlich etc., adsorption kinetics, ). The influence of various factors on the adsorption process is very important.
- Page 8, Line 197; I quote ,,In addition, the physicochemical properties of the ACs are far better than other agricultural materials ….’’ I do not fully agree. Please prove it to be so.

Author Response
Please see the attachment in the box

Round 2
Reviewer 1 Report
the manuscript is now acceptable for publication since the author has reviewed it correctly
Reviewer 2 Report
The manuscript has been improved. I accept corrections.